# Cholesterol Dependent Activity of the Adenosine A_2A_ Receptor Is Modulated via the Cholesterol Consensus Motif

**DOI:** 10.3390/molecules27113529

**Published:** 2022-05-31

**Authors:** Claire McGraw, Kirsten Swonger Koretz, Daniel Oseid, Edward Lyman, Anne Skaja Robinson

**Affiliations:** 1Department of Chemical and Biomolecular Engineering, Tulane University, New Orleans, LA 70118, USA; cmcgraw1@tulane.edu (C.M.); doseid@gmail.com (D.O.); 2Department of Chemical Engineering, Carnegie Mellon University, Pittsburgh, PA 15213, USA; kkoretz@andrew.cmu.edu; 3Department of Physics and Astronomy, University of Delaware, Newark, DE 19711, USA; elyman@udel.edu

**Keywords:** cholesterol, radioligand binding, CGS21680, ZM241386, Gα_s_, surface plasmon resonance, methyl β cyclodextrin

## Abstract

Background: Membrane cholesterol dysregulation has been shown to alter the activity of the adenosine A_2A_ receptor (A_2A_R), a G protein-coupled receptor, thereby implicating cholesterol levels in diseases such as Alzheimer’s and Parkinson’s. A limited number of A_2A_R crystal structures show the receptor interacting with cholesterol, as such molecular simulations are often used to predict cholesterol interaction sites. Methods: Here, we use experimental methods to determine whether a specific interaction between amino acid side chains in the cholesterol consensus motif (CCM) of full length, wild-type human A_2A_R, and cholesterol modulates activity of the receptor by testing the effects of mutational changes on functional consequences, including ligand binding, G protein coupling, and downstream activation of cyclic AMP. Results and conclusions: Our data, taken with previously published studies, support a model of receptor state-dependent binding between cholesterol and the CCM, whereby cholesterol facilitates both G protein coupling and downstream signaling of A_2A_R.

## 1. Introduction

G protein-coupled receptors (GPCRs) comprise the largest known family of human receptors. GPCRs are often therapeutic targets due to their characteristic seven transmembrane domains binding extracellular ligands in order to promote intracellular signaling cascades that can alter gene expression. However, isolation of GPCRs from the cell membrane for further characterization has proven challenging, due in part to tendency of the hydrophilic and hydrophobic segments of the receptor to irreversibly unfold during extraction. For this reason, characterization of the structure-function relationship by crystallography often requires receptor stabilization, and the subsequent biophysical characterization of wild-type receptor can confirm predictions and analysis of structural information.

Adenosine levels in the body have been estimated to lie between 30–200 nM under baseline conditions, but levels can increase in response to cellular damage or stress [1]. Adenosine affects many aspects of cellular physiology, including neuronal activity, vascular function, and blood cell regulation, and mediates its effects mainly through binding to one of the four subtypes of adenosine receptors (ARs), named A_1_, A_2A_, A_2B_, and A_3,_ which are a subfamily of class A GPCRs [2,3]. ARs have also been implicated in neurodegenerative diseases such as AD and Parkinson’s Disease (PD) [4]. Additionally, A_2A_R antagonism has been extensively studied as a therapeutic for PD [5].

One important component of cellular membranes is cholesterol, an isoprenoid-derived lipid that is essential in sustaining structural stability and can also modulate biological processes. Cellular cholesterol levels are highly regulated to ensure proper cell function throughout the body, albeit levels change upon aging [6]. The brain is one of the most cholesterol-rich organs and stores up to 20% of the total cholesterol in the body [7], which is why alterations in cholesterol metabolism are often linked to deficits in brain function. For these reasons, cholesterol dysregulation has been implicated in Alzheimer’s Disease (AD), in addition to other neurodegenerative diseases [8,9,10,11].

Cholesterol is also necessary for the activation of certain membrane proteins, including the adenosine A_2A_ receptor (A_2A_R), a class A GPCR [12,13]. Depletion of cholesterol from the cell membrane significantly reduced downstream signaling of the receptor, as indicated by reduced cAMP levels [12]. Additionally, the in vitro activity of purified receptors was affected by alterations to cholesterol concentrations, as seen by ablation of radioligand binding for A_2A_R purified without cholesterol hemisuccinate [13]. Despite a clear association of cholesterol levels and A_2A_R activity, little is known regarding specific cholesterol binding sites on the receptor. In this work, we created point mutations at predicted cholesterol binding sites to examine the effects of cholesterol association on specific amino acids in A_2A_R.

A specific cholesterol binding site capable of binding two cholesterol molecules was predicted between helices I, II, III, and IV of another class A GPCR, the β_2_ adrenergic receptor (β_2_AR) [14]. This predicted cholesterol binding site established the cholesterol consensus motif (CCM) from one crystal form of the β_2_AR. The CCM is comprised of five highly conserved amino acid residues across the class A subfamily, present in 21% of class members. In class A GPCRs the CCM residues are as follows: [4.39–4.43(R,K)]—[4.50(W,Y)]—4.46(I,L,V)]—[2.41(F,Y)], listed in order of predicted strength of the interaction [14], and using Ballesteros-Weinstein numbering [15]. Following the publication of the β_2_AR structure, Adamian and colleagues suggested the inclusion of 2.45(S) based on bioinformatic analysis [16]. A_2A_R is also predicted to have a CCM [14]; the corresponding residues are Y43(2.41), S47(2.45), K122(4.43), I125(4.46), and W129(4.50) [17]. The tyrosine and lysine residues of the CCM are positioned to form hydrogen bonds with the hydroxyl group in cholesterol, while the isoleucine residue could form hydrophobic contacts with cholesterol. The tryptophan residue is predicted to form a ring stacking interaction with the ring in cholesterol. While the serine residue does not appear to interact with cholesterol directly in crystal structures, it is positioned to form a hydrogen bond with W129 (4.50), the most conserved amino acid within the CCM [17]. In prior publications, the serotonin_1A_ receptor has been identified as a GPCR that interacts with membrane cholesterol. Further experimental work altering membrane cholesterol availability to serotonin_1A_ receptors elucidated the sensitivity of serotonin_1A_ receptors to cholesterol [18,19,20,21,22,23].

Of the 96 GPCRs for which there are crystal structures and a putative CCM, only two contain cholesterol bound in the crystal structure [24]. Even for A_2A_R, there are crystal structures with bound cholesterol, but it is not localized near the CCM. This is likely due to receptor modifications necessary for stabilizing the receptor prior to crystallization and does not necessarily equate to a functional role. For example, one A_2A_R CCM mutation we explore in this work, K122A, is present in an antagonist-favored mutant of A_2A_R, Rant21. Rant21 is a C-terminally truncated A_2A_R variant (Δ316) with five point mutations, including K122A, that was designed to favor antagonist binding and have increased thermostabilization for ease of receptor crystallization; however, as at least one of these amino acids may interact with cholesterol, its alteration may change structured cholesterol sites [25]. Alternatively, the cholesterol may only bind nonspecifically to A_2A_R. In one review, Taghon and colleagues found that cholesterol binding motifs such as CCM, cholesterol recognition amino acid consensus motif (CRAC), and CARC, the reverse of the CRAC motif, are not reliably predictive of cholesterol association [24].

Crystal structures, molecular dynamics simulations, and ^19^F NMR studies have given much insight into potential specific interaction sites between cholesterol and A_2A_R [26,27]. [17,28] Several recent high-resolution crystal structures of A_2A_R identified cholesterol interacting at other locations on the protein, suggesting the possibility of multiple loci of interaction on A_2A_R [29,30,31,32], consistent with molecular simulations, which predicted additional cholesterol interaction sites on the inner and outer leaflets of helices 5 and 6 [33]. ^19^F NMR of cholesterol and amino acids 2–317 of A_2A_R (A_2A_Δ317R) suggested cholesterol is an indirect allosteric modulator of A_2A_Δ317R [34]. Huang et al. additionally proposed that cholesterol association weakly shifted the receptor toward an active state, as determined by characterizing GTPase activity of purified heterotrimeric G proteins (G_s_α_short_β_1_γ_2_) binding to A_2A_Δ317R in nanodiscs with 0–13% cholesterol. While these results demonstrate small changes in receptor activity, studies of wildtype (WT) A_2A_R and full-length Gα_s_ protein could produce different results. One previous study of purified WT A_2A_R or A_2A_R truncated at amino acid 316 (A_2A_Δ316R) associating with purified Gα_s_ saw a decrease in receptor-G protein association upon truncation of the A_2A_R C-terminus [35]. This suggests that the A_2A_Δ317R variant could be affecting association to G protein, and, therefore, GTPase. However, despite using truncated proteins and less than native concentrations of cholesterol, the ^19^F NMR results found a modest shift in receptor activity toward active states.

Other recently published results revealed cholesterol preferentially binds to the CCM when the receptor is in the active agonist bound state, while interacting with other amino acids when the receptor was in an inactive antagonist bound state [12]. To better understand the possible interactions between cholesterol and specific amino acid side chains, we created single amino acid mutations within the CCM of A_2A_R and examined the effects of these variants on extracellular ligand binding, intracellular signaling pathways, and bulk cholesterol depletion. Here, we found these variants show cholesterol-dependent activity of A_2A_R, suggesting a specific, local interaction mechanism mediated by the CCM.

## 2. Results

### 2.1. Importance of the Cholesterol Consensus Motif to A_2A_R Ligand Binding

To identify specific interactions between cholesterol and A_2A_R at the CCM site-directed mutagenesis at sites S47, K122, and W129 (Figure 1) was conducted to mutate each amino acid to alanine (Appendix A). These sites were selected to prevent hydrogen bonding and ring stacking interactions between residues of the CCM and/or cholesterol. To experimentally test the effect of the loss of these contacts, all variants were transiently transfected as pCEP4-A_2A_R constructs into HEK293 cells, and expression was quantified by western blot (Appendix A). While slight differences in expression were observed compared to WT A_2A_R, there were no statistically significant changes in expression or membrane integration for the variants, as determined by densitometric analysis. No significant changes to receptor localization were observed when comparing A_2A_R to the CCM variants examined (Appendix A), indicating that variants were trafficked to the plasma membrane to similar extents as WT A_2A_R.

To determine whether the loss of contact with cholesterol at specific amino acids within the CCM resulted in a reduced capacity to bind extracellular ligand, ligand binding of radiolabeled agonist ([^3^H] CGS21680) or antagonist ([^3^H] ZM241386) in membrane preparations of HEK cells was characterized for the S47A, K122A, and W129A variants and compared to WT A_2A_R (Figure 2). The W129A variant led to the greatest reduction in ligand binding capacity (B_max_; Table 1) when compared to A_2A_R for both agonist and antagonist binding. Interestingly, the presence of the W129A mutation caused a slight decrease in affinity for agonist, and a slight increase in affinity for antagonist. Surprisingly, the S47A mutation also led to a reduction in ligand binding capacity of both agonist and antagonist compared to A_2A_R; however, S47A showed a much greater affinity for agonist than A_2A_R. For the K122A variant, a slight increase in B_max_ and a slight decrease in K_D_ were observed for antagonist binding, while a significant decrease in affinity for agonist was observed (Figure 2; Table 1). Previous work by our lab found that cholesterol depletion in whole cells with MβCD was unable to remove enough cholesterol to affect ligand binding of WT A_2A_R [12]; however, these point mutations to the CCM prevent cholesterol association at specific sites on A_2A_R directly, which could explain why we see a change in ligand binding characteristics by this approach.

### 2.2. A_2A_R CCM Variants Affect Downstream G Protein Coupling

To characterize the effects of CCM mutations on the first step of downstream signaling, G protein coupling, purified A_2A_R protein or a CCM variant was flowed across purified Gα_s_ and near real-time association observed by surface plasmon resonance (SPR). WT A_2A_R was found to have a dose-dependent response to G protein association (Figure 3), as seen in previous studies [35]. Experimental conditions for A_2A_R were repeated for the S47A and W129A variants, as these variants showed the most significant ligand binding differences of the variants. We observed similar association to Gα_s_ for the purified S47A variant, where S47A demonstrated dose-dependent binding, as well as wild-type-like equilibrium binding constants (Table 2). The S47A variant has a similar kinetic association rate constant to WT A_2A_R, but a much faster kinetic dissociation rate constant. This result suggests the S47A variant promotes faster G protein turnover, thereby likely leading to an increase in constitutive G protein signaling. The W129A variant ablated binding to Gα_s_, which suggests cholesterol association to W129 is necessary for intracellular A_2A_R signaling.

### 2.3. A_2A_R CCM Variants Affect Downstream Signaling as Measured by cAMP

To observe how receptor and CCM variant association to G protein translated to amplified downstream signaling pathways, cAMP assays were used to measure both constitutive and agonist (1 µM CGS 21680) stimulated cAMP formation (Figure 4). The S47A variant led to an increase in constitutive activity compared to WT A_2A_R, but upon S47A activation by agonist, cAMP levels were not significantly different than WT. This suggests ligand activated S47A behaves similarly to WT A_2A_R. That is, active S47A does not appear to activate increased intracellular signaling; however, the increase in S47A constitutive signaling taken together with a faster k_off_ from Gα_s_ suggests that apo-S47A may behave more similarly to R* (activated) than R (apo, inactive) A_2A_R.

Expression of K122A and W129A variants led to a decrease in constitutive and agonist induced cAMP, with W129A resulting in a greater reduction in cAMP signaling than K122A. This observation suggests that cholesterol association to K122 and W129 is critical for A_2A_R to initiate downstream signaling, as mutating these amino acids to alanine led to a significant reduction in intracellular responses. This result further indicates that S47 is implicated in cholesterol binding to A_2A_R and contributes to regulating constitutive activity but has less of an effect on agonist-activated downstream signaling.

### 2.4. Bulk Cholesterol Depletion Effects on CCM Variant Signaling

To determine whether bulk cholesterol changes would alter A_2A_R function in the context of these variants, bulk cholesterol depletion was carried out by MβCD addition and receptor activity measured by cAMP assay (Figure 5). Our previous studies indicated that the addition of MβCD to cells expressing WT A_2A_R led to nearly a 60% reduction in cholesterol levels, as determined by cAMP formation. However, membrane cholesterol was restored upon addition of cholesterol-loaded MβCD, as cAMP concentrations were similar to cells expressing WT A_2A_R, untreated with MβCD [12]. As membrane cholesterol was depleted from the cells expressing the variants, varying results were observed. S47A was the most sensitive to MβCD treatment and showed a dose-dependent response. In the presence of agonist, the measured cAMP concentration of S47A was reduced by 80% following 5 mM MβCD addition (Figure 5B) treatment as compared to the untreated control. The K122A variant was similarly sensitive to cholesterol removal as WT A_2A_R. At 5 mM MβCD treatment, the measured cAMP concentration for K122A was reduced by 40%, and for WT A_2A_R the decrease was 43% (Figure 5A,C) [12]. These data suggest cholesterol retains some association to both the S47A and K122A variants, as removal of cholesterol results in a significant decrease in receptor activity. The W129A variant was the least sensitive to MβCD cholesterol depletion, as cAMP activity was unaffected at lower concentrations of MβCD, while treatment at 5 mM MβCD reduced measured cAMP concentrations only by 38% (Figure 5D). Although cAMP was significantly reduced at 5 mM MβCD addition, it is important to note that untreated W129A produced less cAMP than WT A_2A_R. Taken together, this suggests cholesterol association at W129 is required to retain wild-type-like downstream signaling activity.

## 3. Discussion

Previously, we have shown the importance of bulk cholesterol levels in supporting A_2A_R function for both in vitro ligand binding and downstream signaling in the mammalian lipid bilayer [12,13]. All-atom simulations revealed cholesterol binds to the CCM of A_2A_R when the receptor was in the active state [12], while ^19^F NMR of cholesterol and amino acids 2-317 of A_2A_R (A_2A_Δ317R) suggested that cholesterol binding had a slight shift to the activated state [34]. To determine whether cholesterol functions through specific interactions with A_2A_R, as opposed to altering bulk membrane properties alone, amino acids within the CCM were mutated to alanine and activity of CCM variants was compared to WT A_2A_R.

The S47A variant had overall modest effects on receptor activity. Radioligand binding to the S47A variant resulted in a decreased B_max_ for both agonist and antagonist but showed an increase in agonist binding affinity. Despite binding less total ligand, the purified S47A variant retained wild-type levels of binding to purified Gα_s_, albeit our data indicates a faster k_off_ from G protein. Additional downstream signaling data, cAMP activation, is an amplified indicator of G protein signaling, and demonstrated an increase in constitutive S47A signaling, but in the presence of agonist, cAMP levels were similar to wild-type A_2A_R. A faster rate of G protein turnover, as determined by SPR binding, could account for the increase in constitutive G protein signaling, as the higher off-rate could lead to more interaction of the GTP-bound Gα subunit with adenylyl cyclase. Taking the radioligand binding data into account suggests that an increase in agonist affinity, despite a decreased B_max_, in concert with faster G protein turnover somewhat negate one another, as agonist induced cAMP levels are similar to wild type. Furthermore, the S47A variant likely retains some cholesterol interaction, as depletion of membrane cholesterol resulted in a decrease in cAMP formation. From crystal structures, S47 appears to form a hydrogen bond with W129; therefore, our data suggests that the disruption of this hydrogen bond affects normal cholesterol association, but modestly alters the receptor activity.

Molecular dynamics simulations showed the 4.39–4.43(R,K) residue (K122A for A_2A_R) of the CCM was predicted to have the strongest interaction with cholesterol [14]. The K122A variant affected radioligand binding by decreasing receptor affinity 2-fold for both agonist and antagonist, as well as modestly increasing antagonist B_max_. This aligns with the predicted effect of this point mutation, as K122A is one component of a thermostabilized, antagonist-favored variant of A_2A_R [25]. In our work, further exploration into the effects on receptor signaling demonstrated a decrease in both constitutive and agonist induced cAMP compared to wild-type A_2A_R, suggesting that preventing cholesterol interaction at K122 negatively effects constitutive signaling, although the decrease in agonist-induced cAMP could be due to either a decrease in agonist binding or effects of reduced cholesterol association at K122. Membrane cholesterol depletion by MβCD demonstrated a similar effect on K122A as wild-type A_2A_R. That is, cAMP concentrations decreased in the presence of 5 mM MβCD, suggesting cholesterol still affects receptor activity when K122 is mutated to alanine.

W129 is the most conserved amino acid within the CCM across class A GPCRs [14], and recently published molecular dynamics simulations revealed a low affinity cholesterol-binding site at W129A [36]. Upon testing the effects of mutating W129 to alanine on radioligand binding, we found that the W129A variant had the greatest decrease in B_max_ of the three variants we tested, a slight decrease in agonist affinity, and a slight increase in antagonist affinity. When characterizing W129A binding to Gα_s_ with purified protein, we found a significant decrease in G protein association. This result aligns with the ablation of constitutive cAMP formation, as G protein association and dissociation are necessary for initiating intracellular signaling cascades. Furthermore, as W129A showed a decreased agonist binding, as well as a modest decreased agonist affinity, the decrease in agonist-induced cAMP suggests cholesterol association to W129 has an overall significant effect on functional states.

One explanation for these observations is that tryptophan is predicted to ring stack with cholesterol, and mutating W129 to alanine would prevent this favorable interaction. We confirmed that even at 5 mM MβCD, the W129A variant was insensitive to depletion of membrane cholesterol, suggesting that the presence of this mutation already caused a functional disruption in the interaction between cholesterol and the receptor.

Another cholesterol binding motif, the cholesterol recognition amino acid consensus (CRAC) motif was determined from the peripheral-type benzodiazepine receptor (PBR), and is defined by the presence of the sequence pattern -L/V-(X)_1–5_-Y-(X)_1–5_-R/K-, where X represents any amino acid [37]. PBR, although not a GPCR, is a five-transmembrane domain mitochondrial translocator protein responsible for transporting cholesterol from the outer to inner mitochondrial membrane. By manual sequence alignment, the CRAC motif was found in three representative human GPCRs – rhodopsin, the β_2_-adrenergic receptor and the serotonin_1A_ receptor, suggesting an additional specific cholesterol-binding site [19]. The CRAC motif is also present in the human A_2A_R from amino acids 191-199, located on helix 5. Previous analysis of crystal structures suggests that the presence of a CCM or CRAC/CARC motifs is unable to reliably predict cholesterol association in GPCRs [24]. However, since GPCRs function dynamically, our experimental results showing cholesterol interactions at predicted CCM amino acids suggests further exploration of cholesterol association to CRAC amino acids within A_2A_R may be merited.

In summary, our work gives important insight into the complex mechanism by which cholesterol modulates A_2A_R function and should have relevance to other class A GPCRs. Our results suggest that cholesterol modulates A_2A_R cAMP activation through specific interactions at the CCM in a state-dependent manner. Overall, this work substantiates the importance of cholesterol in GPCR activation and opens the door to using this knowledge to develop potential therapies to modulate GPCR disease-related pathways, including neurodegenerative diseases. Future experiments will investigate the ligand and downstream signaling dependence of additional putative cholesterol binding sites, to further understand specific interactions between cholesterol and GPCRs.

## 4. Materials and Methods

### 4.1. Materials

Cholesterol and Methyl-β-cyclodextrin (MβCD) were obtained from Sigma-Aldrich (St. Louis, MO, USA). Fetal bovine serum (FBS), Lipofectamine 2000 transfection reagent and Opti-MEM reduced serum media were from Invitrogen Life Technologies (Carlsbad, CA, USA). CGS21680 and NECA were obtained from Tocris (Bristol, UK), and [^3^H] CGS21680 and [3^H^]ZM241385 were obtained from American Radiolabeled Chemicals (St. Louis, MO, USA).

### 4.2. Site-Directed Mutagenesis

Mutagenesis primers (Appendix A) were designed using the Agilent Quikchange primer design program and were purchased from Eurofins genomics (Louisville, KY, USA). Site-directed mutagenesis was performed on pCEP4-A_2A_R using Quikchange II XL (Agilent, Santa Clara, CA, USA) according to the manufacturer’s protocol. Mutated DNA was then transformed in DH5α chemically competent cells, and mutations were verified through DNA sequencing.

### 4.3. Cell Culture

Human embryonic kidney (HEK293) cells were maintained in growth media containing Dulbecco’s modified eagle medium (DMEM; Cellgro, Manassas, VA, USA) with 10% FBS at 37 °C in a 5% CO_2_ incubator.

### 4.4. Lipofectamine Transfection

Cells were seeded on day 0 in a T-25 flask to be approximately 70% confluent. On day 1, cells were transfected using 10 μL Lipofectamine 2000 reagent, and 1 μg DNA in 2 mL Opti-MEM reduced serum media. On day 2 cells were placed back in growth media and used for experimentation on day 3, as previously described [12].

### 4.5. Membrane Preparation from HEK

Transiently transfected HEK293-A_2A_R cells were scraped, pelleted, and resuspended in ice-cold 1X TE buffer (1% 1M Tris-HCl (pH 7.5) 0.2% 500 mM EDTA (pH 8)) with protease inhibitors. Cells were sonicated with a Bronson Sonifier 450 at 50% power for 30 pulses, and then centrifuged at 2000× *g* for 5 min at 4 °C to remove cell debris. The supernatant was then centrifuged at 100,000× *g* for 1 hr at 4 °C to pellet cell membranes. Membranes were solubilized in 1X RIPA buffer with protease inhibitors; if necessary, membranes were sonicated again for five pulses at 50% power to break up any visible pieces of membrane. BCA assay (Pierce; Rockford, IL, USA) was performed to determine the total protein concentration of isolated membrane, using bovine serum albumin (BSA) (23209; Thermo Fisher, Waltham, MA, USA) as a standard, and membrane preparations isolated that day.

### 4.6. Radioligand Binding Assay

Isolated cell membranes were resuspended in ligand binding buffer (50 mM Tris-HCl, 10 mM MgCl2 and 1 mM EDTA, pH, 7.4) and 5 μg membrane protein/well were loaded onto poly(ethyleneimine) (0.1% *v*/*v*) treated 96-well glass fiber filter plates 49 (MultiScreen-FC filter type B, Millipore, Billerica, MA) as previously described [38]. Cells were incubated with 1.25–40 nM [^3^H]ZM241385 in the presence or absence of unlabeled competitor ligand (50 μM NECA) or 1.25-250 nM [^3^H] CGS21680 in the presence or absence of unlabeled competitor ligand (50 μM CGS21680) for 1.5 h. Once binding equilibrium was reached, membranes were washed three times with ice-cold binding buffer and then 30 μL of scintillation solution (ULTIMA gold, Perkin Elmer) was added to each well. Radioactive counts (CPMs) using a Perkin-Elmer 1450 Microbeta liquid scintillation counter were measured to determine ligand binding approximately 24 h after the addition of the scintillation solution. Non-specific binding was determined from binding to membranes in the presence of an unlabeled competitor and CPMs were subtracted from total binding to calculate specific binding.

### 4.7. Receptor Expression and Purification

A_2A_R and variants were expressed and purified as previously described [38]. Briefly, receptors were expressed in yeast strain BJ5464 (*MATa ura3-52 trp1 leu2Δ1 hisΔ200 pep4::HIS3 prb1Δ1.6R can1* GAL) using pITy4, a multi-integrating vector containing a Gal1-10 promoter to induce protein expression by galactose. For the SPR experiments, a His_6_ tag was already expressed on Gα_s_ and used to attach Gα_s_ to the SPR chip. For this reason, A_2A_R and variants were designed to be purified by rho-1d4 tag (TETSQVAPA) to prevent receptor association to the SPR chip.

BJ5464 cells containing pITy4-A_2A_R-1d4 and variants were grown overnight in YPD (1% yeast extract, 2% peptone, 2% glucose) at 30 °C and 275 RPM. Once cultures reached stationary phase, as determined by measuring an OD_600_ >13, one OD_600_ of cells was transferred to an 800 mL flask containing YPG (1% yeast extract, 2% peptone, 2% galactose). The change of available sugar from glucose to galactose induces receptor expression. Flasks were grown for 30 h at 30 °C and 275 RPM, and 100 mL aliquots of culture (approximately 1500 OD_600_) were pelleted by centrifugation at 3000× *g* for 5 min. The supernatant was discarded, and cell pellets were stored at −80 °C until required for protein purification.

A_2A_R and variants were purified as previously described [35,39]. Briefly, cells were thawed on ice and resuspended with 22 mL lysis buffer (50 mM Tris-HCl buffer (pH 8), 10% glycerol, and 300 mM NaCl) and one cOmplete, EDTA-free Protease Inhibitor tablet (Roche). Approximately 10 mL of 0.5 mm zirconia silica beads (BioSpec Products, Bartlesville, OK, USA) were added and cells were lysed by vortexing for 1 min, cooling on ice for 1 min, and repeated for a total of six cycles. The beads and lysed cells were separated by column and the samples were sonicated at 50% power for 20 s, placed on ice for 20 s, and sonicated a second time. Cell debris was removed by centrifugation at 3200x *g* for 30 min at 4 °C. Cell debris pellets were discarded, and cell membranes were pelleted by ultracentrifugation of the supernatant at 100,000× *g* for 1 h. Pelleted membranes were resuspended in lysis buffer containing 0.1% *n*-dodecyl-*β*-d-maltopyranoside (DDM), 0.1% 3-[(3-cholamidopropyl)-dimethylammonio]-1-propane sulfonate (CHAPS), 0.02% cholesterol hemisuccinate (CHS) (Anatrace, Maumee, OH, USA), and one cOmplete Protease Inhibitor tablet. Samples were gently agitated overnight at 4°C. The next day, insoluble proteins were removed by ultracentrifugation at 70,000× *g* for 1 h, and the supernatant was incubated overnight with 0.5 mL Rho-1d4 resin (Cube Biotech, Mannheim, Germany). The next day, the resin was washed three times with 15 mL of wash buffer (lysis buffer containing 0.1% DDM, 0.1% CHAPS, 0.02% CHS, and one cOmplete Protease Inhibitor tablet) to remove any non-specifically bound proteins. To elute proteins, A_2A_R and variants were incubated at 4 °C for 2 h in 2.7 mL of wash buffer with 200 µM Rho-1d4 peptide (Cube Biotech). Samples were desalted in PD-10 desalting columns (GE Healthcare, Chicago, IL, USA) previously equilibrated with wash buffer. Receptor concentrations were determined by A_280_ measurement (ε1% = 12.0). Purified receptors were stored at 4 °C and used within 1 week of purification.

### 4.8. G Protein Expression and Purification

Gα_s_ was expressed and purified as previously described [35]. Briefly, pET15b-Gα_s_ construct was transformed into Rosetta (DE3) cells and grown on an LB-Amp-Cam plate (1% tryptone, 0.5% yeast extract, 1% NaCl, 100 µg/mL ampicillin, and 25 µg/mL chloramphenicol). Individual colonies were selected to inoculate 10 mL culture tubes of LB-Amp-Cam and grown at 37 °C and 250 RPM for 12 h. When media reached an optical density at 600 nm (OD_600_) ~10, 20 mL of culture was added to 1 L of LB-Amp-Cam media and grown at 30 °C and 250 RPM until OD_600_ reached 0.6, about 2–3 h. A total of 50 µM IPTG was added to induce expression of Gα_s_ protein, and flasks were grown at 30 °C and 250 RPM for 12–15 h. Cells were pelleted by centrifugation at 10000x *g* and stored at −80 °C until needed for purification.

Gα_s_ was purified from frozen cell pellets as previously described [35], by binding to Ni-NTA resin. Eluted Gα_s_ protein (~50 mLs from 1L of cells) was concentrated to ~2 mL via Amicon Ultra-15 10K Centrifugal Filter Devices (MilliporeSigma, Burlington, MA, USA), and buffer exchange was performed with ~25 mL dilution buffer (50 mM Tris-HCl (pH 8), 1 mM dithiothreitol, and 10% glycerol) to remove residual imidazole and EDTA. The buffer exchanged purified sample was concentrated to ~2 mL via Amicon Ultra-15 10K Centrifugal Filter Device (10–30 mg/mL final concentration).

### 4.9. Surface Plasmon Resonance (SPR) 

In vitro interactions between A_2A_R variants and Gα_s_ were characterized by 4-channel SPR (Reichert, Inc., Depew, NY, USA). All SPR experiments were conducted at 20 °C and a flow rate of 25 µL/min with a high-capacity Ni-NTA chip (Xantec, Duesseldorf, Germany) to bind purified Gα_s_ protein, and running buffer containing detergents (50 mM Tris-HCl (pH 8), 0.1% DDM, 0.1% CHAPS, 0.02% CHS). Detergents were selected at concentrations well above the critical micelle concentration (CMC) so micelles would form and maintain proper receptor activity. This running buffer was used for the duration of SPR experiments instead of just during receptor interactions to minimize the bulk shift during sample injections.

SPR experiments were conducted on a four-channel sensor chip, such that active G protein was attached to channel one, and denatured G protein was attached to channel two to act as a negative control. Prior to attaching G protein, 40 mM NiCl_2_ was injected across all channels for 3 min to activate the NTA chip with nickel in preparation for His_6_ binding. Next, 1 µM of purified, active Gα_s_ was injected across channel one for 3 min and dissociation was observed for 5 min. This was repeated for purified, denatured Gα_s_ on an alternate channel. Typical µRIU responses were at least 300 µRIU, indicating G protein was successfully bound to the appropriate channel.

Following attachment of active and denatured G protein to separate channels, purified A_2A_R or variants were injected across all channels at 296, 445, or 667 nM concentrations for 3 min and dissociation observed for 5 min. A total of 350 mM EDTA was then injected for 30 s across all channels to chelate the bound nickel, thereby removing both the nickel and proteins from the NTA chip. Next, 20 mM NaOH was injected for 1 min to ensure complete removal of bound proteins and reset the chip for subsequent experiments. Prior to moving forward with the stated method, several conditions were tested to ensure sufficient attachment of purified G protein to chip, that there were no mass transfer limitations during protein-protein association, and that regeneration conditions fully removed bound protein from the sensor chip.

### 4.10. SPR Analysis

Specific binding of purified receptor to purified G protein was calculated by subtracting nonspecific binding of receptor to denatured G protein (e.g., channel 2) from total binding of receptor to active G protein (e.g., channel 1). After determining the specific binding curves, data was exported to TraceDrawer software (Ridgeview Instruments, Uppsala, Sweden) and curves were fitted per purification (per biological replicate). That is, duplicates for one biological replicate at all three concentrations were used to fit kinetic binding curves (e.g., purified A_2A_R at 296, 445, and 667 nM) using a 1-1 binding model (Appendix A). Then, fits for each of the three biological replicates were averaged to determine equilibrium and kinetic rate constants.

### 4.11. cAMP Activity Assay

HEK cells expressing A_2A_R and CCM variants were incubated for 30 min in the presence or absence of 1 μM CGS21680 at a cell density of 1,000 cells/well in a white 384 well plate (Grenier bio-one #784075, Kremsmünster, Austria). The concentration of cAMP per well was measured using the cAMP dynamic 2 kit (CisBio, Bedford, MA, USA) using a BioTek Synergy H1 Plate Reader according to manufacturer’s protocol.

### 4.12. Cholesterol Depletion by MβCD of Cells in Culture

HEK cells expressing CCM variants were depleted of cholesterol using 1.25 mM to 5 mM methyl-β-cyclodextrin (MβCD) for 30 min at 37 °C as previously described [12].

### 4.13. Statistical Analysis

Prism 9.3 (GraphPad software, L.L.C., San Diego, CA, USA) was used for student’s *t*-test analysis. Values were considered statistically significant if *p* < 0.05.

## Figures and Tables

**Figure 1 molecules-27-03529-f001:**
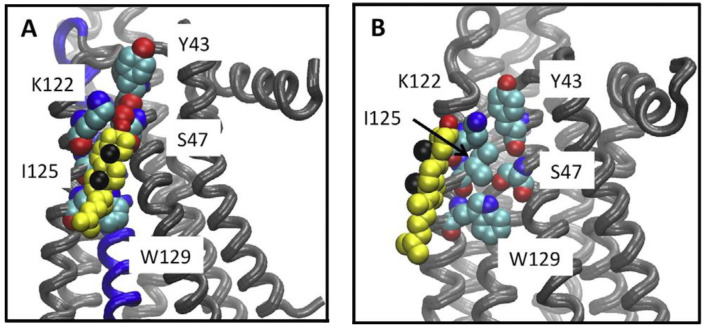
Cholesterol association to the A_2A_R CCM. Cholesterol bound to the CCM during simulations. (**A**) A snapshot from the UK432097-bound simulation, showing the disposition of cholesterol (yellow, black spheres show the methyls of the beta face and red the hydroxyl) relative to the CCM (shown in space filling representation) in a tightly bound configuration. (**B**) The closest approach obtained between cholesterol and the CCM in any of the inactive receptor simulations (snapshot from ca. 0.8 μsec of the ZM241385 bound simulation. Figure is adapted reprinted with permission (license #5316530596446) from McGraw et al. [12]. Copyright 2019, Elsevier B.V.

**Figure 2 molecules-27-03529-f002:**
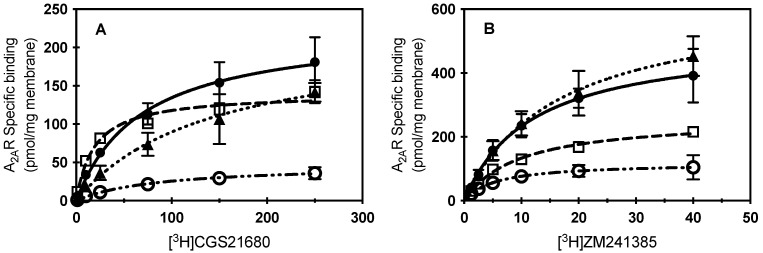
Specific ligand binding of CCM A_2A_R variants. (**A**) Specific binding of increasing concentrations of [^3^H] CGS21680 to membrane preparations from cells expressing WT A_2A_R (filled circles), S47A (open squares), K122A (filled triangles), or W129A (open circles). Mean ± S.D. values from *n* ≥ 2 experiments performed in duplicate. (**B**) Specific binding of increasing concentrations of [^3^H]ZM241385 to membrane preparations from cells expressing WT A_2A_R (filled circles), S47A (open squares), K122A (filled triangles), or W129A (open circles). Mean ± S.D. values from *n* ≥ 2 experiments performed in duplicate. Lines indicate the fit of a one-one binding model for the data (kinetic parameters listed in Table 1; WT A_2A_R, solid line; S47A, dashed; K122A, dotted; or W129A, dash-dotted.

**Figure 3 molecules-27-03529-f003:**
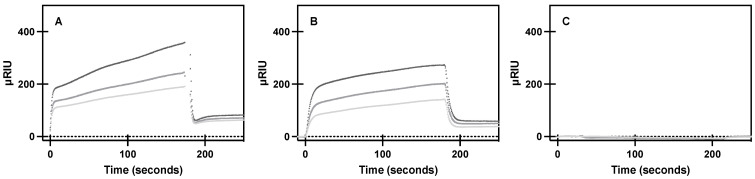
Specific binding of purified A_2A_R and variants to purified Gα_s_. SPR sensorgram curves show specific association and dissociation of 267, 445, and 667 nM (light gray, medium gray, dark gray, respectively) of purified (**A**) A_2A_R, (**B**) S47A, and (**C**) W129A to purified Gα_s_ bound to a Ni-NTA sensor chip, as described in materials and methods. Mean ± S.D. from *n* = 6 (duplicates of 3 biological replicates).

**Figure 4 molecules-27-03529-f004:**
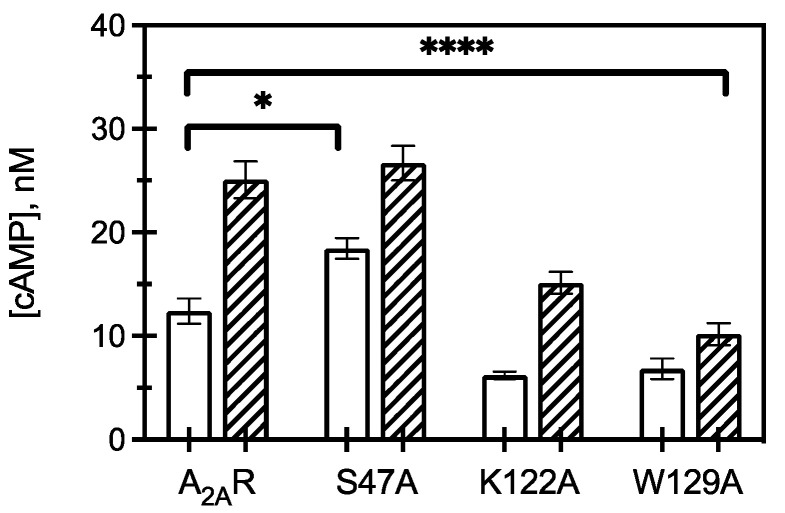
Downstream signaling of CCM variants. cAMP concentrations measured from HEK293 cells expressing A_2A_R or an A_2A_R variant (as listed) in the presence (hatched) and absence (open) of 1 μM CGS21680 (a selective agonist for A_2A_R). The bars represent the mean ± S.E.M. from *n* ≥ 3 independent experiments performed in triplicate. * *p* < 0.05 and **** *p* < 0.0001 are significantly different from A_2A_R control for ligand activated CCM variants.

**Figure 5 molecules-27-03529-f005:**
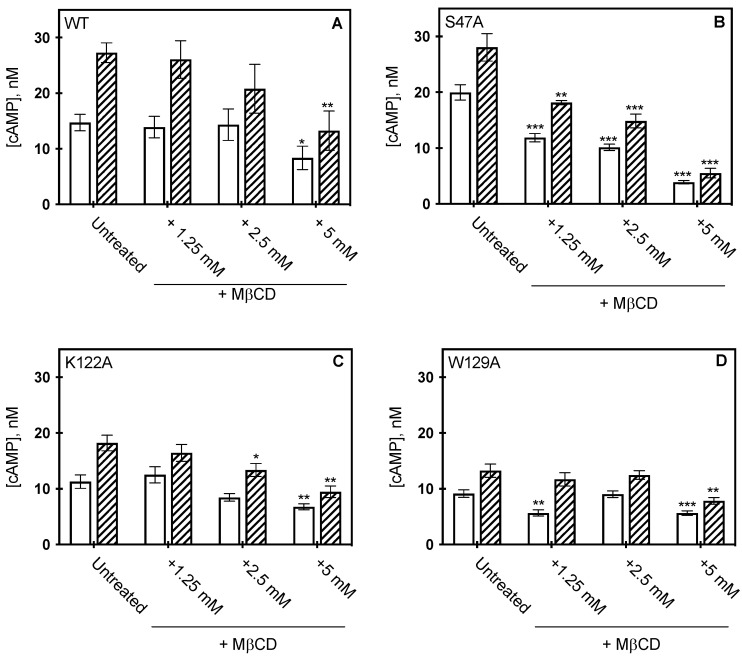
Effect of MβCD on CCM variants. cAMP concentration measured from HEK293 cells expressing wild type A_2A_R (**A**) S47A (**B**) K122A (**C**) and W129A (**D**) following MβCD addition, in the presence (hatched) or absence (open) of 1μM CGS21860 (a selective agonist for A_2A_R). cAMP was measured as described in materials in methods, where the bars represent the mean ± S.E.M error bars from *n* = 3 experiments performed in triplicate. * *p* < 0.05, ** *p* < 0.005, and *** *p* < 0.0005 are significantly different from the untreated control for MβCD addition.

**Table 1 molecules-27-03529-t001:** Ligand binding parameter fits for a one-one model of receptor to ligand. K_D_ and B_max_ values reported as best fit ± S.E. from data shown in Figure 2. * Indicates *p* < 0.01, ** indicates *p* < 0.001, *** indicates *p* = 0.0001.

	CGS21680 K_D_ (nM)	CGS21680 B_max_ (nM)	ZM241385 K_D_ (nM)	ZM241385 B_max_ (nM)
WT A_2A_R	69.0 ± 17.9	227.2 ± 21.3	11.8 ± 2.6	509.7 ± 45.1
S47A	19.1 ± 3.7 *	140.3 ± 6.5 *	9.5 ± 1.1 **	260.3 ± 11.8
K122A	141.2 ± 52.3	216.4 ± 38.3	16.4 ± 3.9	631.7 ± 68.0
W129A	84.9 ± 25.4 ***	47.3 ± 5.5	5.6 ± 1.6 ***	119.1 ± 11.25

**Table 2 molecules-27-03529-t002:** Kinetic parameters determined from a one-one model of binding of purified receptor association to purified Gα_s_ protein via SPR data shown in Figure 3. K_D_ and B_max_ values reported as best fit ± S.E. * Indicates *p* = 0.002.

	k_on_ (1/Ms)	k_off_ (1/s)	K_D_ (nM)	B_max_ (µRIU)
WT A_2A_R	1.27 × 10^4^ ± 649	0.00066 ± 9.3 × 10^−5^	74	161 ± 8.8
S47A	1.87 × 10^4^ ± 2900	0.00122 ± 9.7 × 10^−5^ *	146	133 ± 14.4

## Data Availability

The data presented in this study are available in Appendix A. Any additional information will be provided by the authors upon request.

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
