# Peer review of "Cholesterol Dependent Activity of the Adenosine A2A Receptor Is Modulated via the Cholesterol Consensus Motif"

_molecules, 2022, doi:10.3390/molecules27113529_

Round 1

Reviewer 1 Report

This work is well organized and written. The authors' findings in this work support what they found in the previous work and generalize the cholesterol-dependent activity of the Adenosine A2AR is modulated via the CCM. 

This work is about the importance of cholesterol in the modulation of adenosine A2A Receptor expressed on the membrane of mammalian cells. The manuscript is well organized and written. The authors' findings in this work support what they found in the previous work and generalize the cholesterol-dependent activity of the Adenosine A2AR is modulated via the CCM. In detail, the authors tried to study the cholesterol-dependent activity of the adenosine A2AR with various cell types having mutations on the receptors, but surprisingly the tendency is consistent. Subsequently, it was also confirmed that the resultant cAMP is decreased when MbCD concentration is increased. This manuscript is well deserved for publishing in Molecules.

Author Response

The authors would like to thank the reviewers for the thoughtful and critical comments to improve our manuscript. The manuscript has been carefully revised and the changes are highlighted in RED in the main text.  The reviewers’ comments are reproduced below, and our point-by-point responses are in RED.

Reviewer #1

This work is well organized and written. The authors' findings in this work support what they found in the previous work and generalize the cholesterol-dependent activity of the Adenosine A2AR is modulated via the CCM. 

This work is about the importance of cholesterol in the modulation of adenosine A2A Receptor expressed on the membrane of mammalian cells. The manuscript is well organized and written. The authors' findings in this work support what they found in the previous work and generalize the cholesterol-dependent activity of the Adenosine A2AR is modulated via the CCM. In detail, the authors tried to study the cholesterol-dependent activity of the adenosine A2AR with various cell types having mutations on the receptors, but surprisingly the tendency is consistent. Subsequently, it was also confirmed that the resultant cAMP is decreased when MbCD concentration is increased. This manuscript is well deserved for publishing in Molecules.

Response: We thank the reviewer for their support and feedback.

Reviewer 2 Report

In this manuscript by McGraw et al, the authors investigate the importance of three amino acids in the conserved cholesterol consensus motif (CCM) of the adenosine A2A receptor (A2AR), using mutations to alanine in each of these residues (S47, K122 and W129). The authors show that the mutant forms of A2AR, as expressed in HEK 293 cells, show normal plasma membrane localization, but altered ligand binding, each dependent on the type of mutation and whether they used an agonist or antagonist. When combined with KD and Bmax values, the data indicate that the S47A mutation showed increased affinity for agonist, but a decrease for K122. Interestingly the W129A mutation caused the greatest decrease in ligand binding. Next studies of G protein coupling indicated that the S47A mutation exhibited dose-depending binding similar to the wild-type protein, but this was absent in the W129 variant, suggesting this amino acid is critical to cholesterol binding. Assays for cAMP formation, without (e.g., constitutive) vs. agonist-simulated, showed that S47A mutation caused increased constitutive but not agonist-induced cAMP formation. In contrast, both K122A and W129A mutations decreased both constitutive and agonist-induced cAMP. Together the data indicate that cholesterol binding at the CCM domain in A2AR is important to its capacities to regulate G-protein signaling, and adds to our knowledge of the functional domains of GPCR that are controlled by cholesterol. The manuscript is straightforward with the experimental approach and data presentation, easy to follow and understand, and provides important new data regarding the importance of cholesterol to G-protein coupled receptor functions. There are some minor issues to consider as follows, but no major problems are noted.

Lines 45-54, the authors should add a summary statement as to the biologic importance of A2AR, as they introduce the importance of cholesterol binding to this GPCR. This could then be used for further comments in the Discussion section (e.g., with the summary statements in lines 316-324).

Line 85, the authors should define "CRAC" and "CARC", as they did for "CCM" (similar to that used in the Discussion, line 303).

Line 218, the authors should add after "cholesterol levels dropped by almost 60%" that this refers to the MbCD-treated cells, which can then lead to the statement of restored cholesterol in the cholesterol-loaded MbCD.

Line 228, edit the line "results leads to...", perhaps "results in a significant decrease..."

Author Response

The authors would like to thank the reviewers for the thoughtful and critical comments to improve our manuscript. The manuscript has been carefully revised and the changes are highlighted in RED in the main text.  The reviewers’ comments are reproduced below, and our point-by-point responses are in RED.

Reviewer #2

In this manuscript by McGraw et al, the authors investigate the importance of three amino acids in the conserved cholesterol consensus motif (CCM) of the adenosine A2A receptor (A2AR), using mutations to alanine in each of these residues (S47, K122 and W129). The authors show that the mutant forms of A2AR, as expressed in HEK 293 cells, show normal plasma membrane localization, but altered ligand binding, each dependent on the type of mutation and whether they used an agonist or antagonist. When combined with KD and Bmax values, the data indicate that the S47A mutation showed increased affinity for agonist, but a decrease for K122. Interestingly the W129A mutation caused the greatest decrease in ligand binding. Next studies of G protein coupling indicated that the S47A mutation exhibited dose-depending binding similar to the wild-type protein, but this was absent in the W129 variant, suggesting this amino acid is critical to cholesterol binding. Assays for cAMP formation, without (e.g., constitutive) vs. agonist-simulated, showed that S47A mutation caused increased constitutive but not agonist-induced cAMP formation. In contrast, both K122A and W129A mutations decreased both constitutive and agonist-induced cAMP. Together the data indicate that cholesterol binding at the CCM domain in A2AR is important to its capacities to regulate G-protein signaling, and adds to our knowledge of the functional domains of GPCR that are controlled by cholesterol. The manuscript is straightforward with the experimental approach and data presentation, easy to follow and understand, and provides important new data regarding the importance of cholesterol to G-protein coupled receptor functions. There are some minor issues to consider as follows, but no major problems are noted.

Lines 45-54, the authors should add a summary statement as to the biologic importance of A2AR, as they introduce the importance of cholesterol binding to this GPCR. This could then be used for further comments in the Discussion section (e.g., with the summary statements in lines 316-324).

Response: We have added a paragraph on physiological and pathological roles of adenosine receptors (lines 38-45)

Line 85, the authors should define "CRAC" and "CARC", as they did for "CCM" (similar to that used in the Discussion, line 303).

Response: CRAC and CARC definitions have been added to line 85.

Line 218, the authors should add after "cholesterol levels dropped by almost 60%" that this refers to the MbCD-treated cells, which can then lead to the statement of restored cholesterol in the cholesterol-loaded MbCD.

Response: This text has been edited to clarify the results from the MBCD experiments (now at lines 228-232).

Line 228, edit the line "results leads to...", perhaps "results in a significant decrease..."

Response: The text has been modified as suggested.